# Chimeric Claudins: A New Tool to Study Tight Junction Structure and Function

**DOI:** 10.3390/ijms22094947

**Published:** 2021-05-06

**Authors:** Abigail Taylor, Mark Warner, Christopher Mendoza, Calvin Memmott, Tom LeCheminant, Sara Bailey, Colter Christensen, Julie Keller, Arminda Suli, Dario Mizrachi

**Affiliations:** Department of Physiology and Developmental Biology, College of Life Sciences, Brigham Young University, Provo, UT 84602, USA; abigailelizabethtaylor@gmail.com (A.T.); markjrwarner@gmail.com (M.W.); cmendoz1@vols.utk.edu (C.M.); bigcmemmott@gmail.com (C.M.); tlechem@gmail.com (T.L.); sarambailey21@gmail.com (S.B.); colter47@hotmail.com (C.C.); julieellenkeller@gmail.com (J.K.); as227@byu.edu (A.S.)

**Keywords:** claudin, occludin, tight junction, membrane protein, E-cadherin

## Abstract

The tight junction (TJ) is a structure composed of multiple proteins, both cytosolic and membranal, responsible for cell–cell adhesion in polarized endothelium and epithelium. The TJ is intimately connected to the cytoskeleton and plays a role in development and homeostasis. Among the TJ’s membrane proteins, claudins (CLDNs) are key to establishing blood–tissue barriers that protect organismal physiology. Recently, several crystal structures have been reported for detergent extracted recombinant CLDNs. These structural advances lack direct evidence to support quaternary structure of CLDNs. In this article, we have employed protein-engineering principles to create detergent-independent chimeric CLDNs, a combination of a 4-helix bundle soluble monomeric protein (PDB ID: 2jua) and the apical—50% of human CLDN1, the extracellular domain that is responsible for cell–cell adhesion. Maltose-binding protein-fused chimeric CLDNs (MBP-CCs) used in this study are soluble proteins that retain structural and functional aspects of native CLDNs. Here, we report the biophysical characterization of the structure and function of MBP-CCs. MBP-fused epithelial cadherin (MBP-eCAD) is used as a control and point of comparison of a well-characterized cell-adhesion molecule. Our synthetic strategy may benefit other families of 4-α-helix membrane proteins, including tetraspanins, connexins, pannexins, innexins, and more.

## 1. Introduction

The tight junction (TJ) is the most apical cell adhesion complex, found in epithelial and endothelial tissue [1]. It serves primarily as a barrier, selectively regulating the paracellular passage of solutes based on size and charge [2,3]. The TJ is also involved in cell signaling pathways that control cell motility, apicobasal polarity determination, and vesicular trafficking [3], and it has been found to reside in diverse compartments during the cell cycle, playing key roles regulating transcription and proliferation [4]. A variety of membrane and soluble proteins are integral components of the TJ. Important TJ transmembrane proteins include claudins (CLDNs), immunoglobulin-like proteins (junctional adhesion molecules, JAMs), and TJ-associated MARVEL-domain proteins: occludin (OCLN), tricellulin (TRCL), and MARVELD3 [4]. The cytoplasmic counterpart consists of *zonula occludens* proteins (ZO-1, ZO-2, ZO-3) and protein complexes related to cell polarity determination [5,6]. Previous reviews outline the various structural and functional features of TJ proteins [6,7,8]. Although much is known about the structure and function of TJs [9,10,11], additional research is necessary to construct a cohesive view of the TJ as a whole, as well as to understand its assembly.

The CLDN protein family (human genome consists of 23 annotated genes) is a critical element within TJs [12]. In contrast to OCLN and JAMs, CLDNs are essential for the TJ’s control of the paracellular space [6,13,14]. Aside from its strict function in blood–tissue barriers, CLDNs are thought to play a major role in regulating paracellular ion conductance by forming aqueous pores that regulate the diffusion of ions through the paracellular transport pathway [14]. CLDNs also play a role in determining both apicobasal and planar polarity [15,16,17]. CLDNs are critical in embryonic development [11,16,18,19,20]. Due to their role in the regulation of the tightness of the paracellular barrier, CLDNs, specifically CLDN5, can also regulate hydrostatic pressure within the embryonic lumen, allowing for lumen expansion and brain morphogenesis [21]. More specifically, CLDNs are required in renal branching morphogenesis [22].

To characterize the TJ’s structure, researchers have turned to electron microscopy, as a means to define its apical position and adhesive properties [23]. To study individual membrane proteins of the TJ, multiple strategies have been employed. To study JAMs, for example, the single transmembrane helix was removed in order to crystalize the extracellular domain. The resulting crystal structure of JAM-A was demonstrated to form a homodimer involving the first immunoglobulin domain of the extracellular region [24]. The crystal structure of the C-terminal domains of TJ-associated MARVEL-domain proteins OCLN [9]. and TRCL [25]. are available, giving a small glimpse to intracellular events in the TJ. Full-length CLDNs have been crystallized from detergent-solubilized recombinant proteins [10]. All the structures reported so far depict a monomeric structure although structural information has been extracted [10] and further used for in silico studies [26] in order to learn more of the TJ assembly, structure and function. Detergents are limited in their ability to maintain the natural oligomeric states of membrane proteins [27,28]. Thus, for TJs, and other transmembrane proteins forming quaternary complexes, it is imperative to develop true detergent-free methods for structural characterization [29,30]. Crystal structures of proteins with large fusion proteins, such as maltose-binding protein (MBP), thioredoxin (TRX), or glutathione-S-transferase (GST) or other chimeric designs, have been obtained [31]. For example, chimeric designs have been employed in the crystal structure of G-protein coupled receptors (GPCRs) [32] with significant impact on drug discovery [33]. A prerequisite of experiments such as interaction studies, structural and biophysical characterization, where purified proteins are required, is access to high-quality samples in order to ensure reproducibility and reliability of the data. High quality has been defined as a sample of high-purity, structurally homogeneity where the synthesis thereof is highly reproducible [30].

To generate high-quality samples, researchers design chimeric proteins. In the classic approach, there are two types of chimeric designs for proteins. The first type, the translational 3′-terminus of the first gene is deleted, as is the promoter of the 5′-terminus of the second gene of interest. The two genes are then ligated in-frame, with or without a connecting linker. This type of fusion protein is typically multifunctional, in which each of the fusion partners adds its own separate function (e.g., binding activity, enzymatic activity, improved pharmacokinetic properties, improved solubility and folding characteristics). The crystal structure of MBP fused to PigG, an acyl carrier protein related to the synthesis of prodigiosin, a tripyrrole red pigment, is an example of this first design of chimeric proteins [34]. This type of fusion protein typically contains only a phenotypic activity rather than two separate activities as noted above for MBP-PigG, where MBP provides improved solubility and PigG represents the enzymatic activity [34]. The conclusions of this study indicate that PigG, and a conserved amino acid, are essential for the synthesis of prodigiosin [34]. Worth noting is that both MBP and PigG, although fused in the crystal structure, fold independently of each other [34].

In the second type of chimeric protein, the sequences of two structurally similar proteins are combined in an intertwined fashion to yield a novel protein. In this case, the structural properties are preserved, enabling the conservation of function. The numbers of possible permutations enhance this platform’s versatility and value as a tool for many areas of research and biotechnology. Examples include fusion proteins for half-life extension of biologics [35]. Glucagon-like peptide 1 (GLP-1) fusion chimeras with the constant region of human IgG_2_ (hIgG_2_ F_c_) results in a potent long-acting receptor agonist^36^. Wang and colleagues report that GLP-1/hIgG_2_ F_c_ was effective in reducing the occurrence of diabetes in streptozotocin-induced type 1 diabetes in mice [36].

In the case of CLDNs [10], detergent-based methods have produced only monomeric forms. Nevertheless, researchers have extracted structural information from these experiments—for example, the secondary structure of CLDNs as well as the membrane boundaries [37]. Furthermore, in the case of human CLDN9, researchers have derived an open and closed conformation by crystallizing the protein in complex with *Clostridium perfringens* enterotoxin C-terminal domain (CPE) [38], a well-known CLDN binder [39]. The basis for the lack of Cis- and Trans-interactions could be due to use of detergents appearing to disrupt key protein–lipid or protein–protein interactions fostered within the microenvironment of the membrane [40,41]. CLDN assembly in strands has been studied in silico (reviewed by Fuladi and colleagues [26]). Thus, the tertiary and quaternary structure of CLDNs remain theoretical, based almost entirely on in silico modeling [26,42,43]. In particular, Fuladi et al. propose that some of the in silico models are not compatible with experimental studies [26]. For example, Fuladi cites discrepancies in the role of CLDN’s transmembrane helices in strand formation through *Cis* interactions [26].

The search for modulators of protein–protein interactions (PPIs) is currently a promising strategy leading toward next-generation drugs. This approach remains challenging considering that PPIs of interest generally involve rather flat and large protein areas [44]. In the crystal structure of mouse CLDN15, roughly 50% of the extracellular loops are β-sheet structured [37]. Translational solutions to either overcome transiently the TJ for drug delivery or to strengthen the paracellular pathway are goals unfulfilled by current pharmacological or biotechnological tools. Peptide-based strategies have been used to extract functional structures from CLDN and TRCL extracellular loops [45,46,47]. Unfortunately, these peptides remain as research tools only. Understanding CLDN structure and TJ assembly will be key steps for drug discovery [48] and translational solutions.

In this current study, we present an alternative approach to preserve the structure-function properties of CLDNs through the use chimeric protein strategies. Our design is detergent-independent and produces high-quality samples for structural and functional studies. The strategy presented here is a novel way of combining chimeric strategies with membrane proteins and could be useful for the development of similar approaches for other membrane protein families.

## 2. Results and Discussion

### 2.1. Chimeric CLDN1 Design and Properties

Our first design for a detergent-independent chimeric CLDN sought to replace part of the transmembrane domain of human CLDN1. We examined the transmembrane boundaries reported in the crystal structure of mouse CLDN15 [37] to predict similar boundaries in hCLDN1. We obtained a 3-D homology model of hCLDN1 from Dr. Shikha Nagidi (Department of Biomedical and Chemical Engineering, Syracuse University, NY, USA) [49,50] and used it alongside a primary amino acid sequence alignment to perfect the design. Furthermore, we used a synthetic protein to replace part of the 4-helix transmembrane domains in hCLDN1. Hetch and colleagues designed de novo proteins to produce 4-helix bundles [51]. These proteins are soluble and are deprived of PPIs [52], making them suitable counterparts for CLDNs that are membrane proteins with extensive cis- and trans-interactions. We chose the protein S-836 (PDB id: 2jua) for synthesizing the interwoven chimeric DNA sequence. In Figure 1A, we present the amino acid sequence of chimeric CLDN1, containing 50% of the original protein, and 2jua (CC1). Figure 1B has a graphical representation of CC1.

The CC1 construct was not well behaved in recombinant expression experiments, displaying toxicity to *E. coli*. Yields of CC1 were minimal and unsuitable for downstream applications. We then considered expanding the extent of the chimeric approach by using MBP as a fusion protein containing a short linker leading to CC1. For structural determinations using MBP as a fusion partner, a short linker (Asn-Ala-Ala-Ala) is typically used [53]. Appendix A contains the amino acid sequence of the final construct MBP-CC1. Figure 1C is a graphical representation of MBP-CC1 displaying modeled dimensions (UCSF Chimera) for the combined domains.

The recombinant expression of MBP-CC1 decreased cellular toxicity during growth and following IPTG-induced protein expression. We used a two-step purification scheme, Amylose resin followed by size-exclusion chromatography (SEC). Typical yields of highly purified protein (>95%) were recorded between 5 and 7 mg/L of culture. An alternative design was tested in which cleavage site (TEV protease) between MBP and CC1 yielded mixtures of cut and uncut species that hindered structural studies due to lack of homogeneity of the sample [30]. Considering the MBP-CC1 large oligomeric state, we opted to preserve the homogeneity of the purified species; therefore, MBP is part of all the constructs in this study for consistency.

SEC experiments (Figure 1D) identify MBP-CC1 mostly as a single peak with a size greater than 670 kDa, according to the size standards (see Materials and Methods). Considering the molecular weight of MBP-CC1 (65.3 kDa), the results suggest the formation of quaternary structures of at least a decamer. Even though SEC is widely utilized for initial characterization of protein size, some limitations include the lack of appropriate standard to complete the analysis. SEC is a more qualitative technique leading to an inaccurate calculation of protein size, specifically in extreme behaviors [54]. In order to obtain a more precise estimate of the molecular weight of MBP-CC1, we tested our samples using Dynamic Light Scattering. Unfortunately, DLS, was unreliable when following the manufacturers guidelines to perform measurements (Brookhaven 90Plus Particle Size Analyzer; Brookhaven Instruments Co., Holtsville, NY, USA). DLS has a limit of ~10,000 Å and the failure of the experimentation could be an important indication of the large size of the quaternary assembly of MBP-CC1.

Protein structure conforms to the physical chemistry of its component polypeptide primary sequence interacting with itself and with the surrounding solvent to achieve the most energetically favorable fold. Our data suggest that preserving 50% of hCLDN1 in the chimeric design with 2jua conserved sufficient structural features of CLDNs to foster the minimum amount of energy required for the folding of extracellular loops leading to CLDN–CLDN interactions.

To further understand if the oligomeric forms of MBP-CC1, we resourced to structural studies. Small-angle X-ray scattering for biological molecules (bioSAXS) is an established method for structural characterization of samples at resolutions between 1 nm and 1000 nm [55,56]. When MBP-CC1 was analyzed with bioSAXS, the structure observed (as a volume) reflected a structure of multiple dimensions (Figure 1E). At its lengthiest edge, the structure was ~255 Å with ~145 Å height. When rotated 90°, we observed that the depth of the structure reached 72 Å at the bottom and at the top, with a decreased length (55 Å) at the center of the rendered volume. Radius of gyration, the value that expresses the distribution of the volume around the mass center is in agreement with rod-like structures of large size [56,57,58]. In the bioSAXS experiment, the maximum size of the particles (D_max_), 305 Å, is also in agreement with the dimensions of the structure presented in Figure 1E. The models derived in Figure 1C for MBP-CC1 were used to estimate an organization to the quaternary ultrastructure of the chimera. We propose that under the described conditions, MBP-CC1 organizes as rows of dimers that combine with a second row of dimers through the extracellular loops. *Cis* interactions are responsible for preserving the rows. Our data do not explain if the extracellular domains of CLDN1 alone are sufficient for these interactions or if a combination of the hydrophobic domains also contribute and to what extent. CLDN dimers, organized as strands, have been reported in the literature as basic units of the TJ [59,60]. Zhao et al. proposed that multiple CLDN–CLDN *Cis* interfaces are important for TJ strand formation and confer flexibility [60]. Our findings are consistent with the literature and validates the structure–function preservation in the MBP-CC1 design. Integral membrane proteins fold into their active conformations in a complex milieu dictated by the lipids of a bilayer cell membrane [41,61]. Our chimeric design of a structural 4-helix bundle may be applied to other membrane proteins of similar topology.

### 2.2. Surface Plasmon Resonance (SPR) to Determine MBP-CC1 Constant of Affinity

SPR is an optical technique that can be utilized to measure the binding of proteins in real time without the use of labels. Experimentally, the constants of association (K_a_) and dissociation (K_d_) are obtained. The most common datum reported through SPR is the affinity constant (K_D_) between ligand and analyte. To the best of our understanding, direct PPI measurements for CLDNs are not currently available in the literature. To establish the quality of the MBP-CC1 sample to determine these kinetic values, we employed OpenSPR (Nicoya Life, Kitchener, ON, Canada), a localized SPR (LSPR) where the sensor is coated with gold nanoparticles instead of a continuous sheet of gold [62]. LSPR produces a strong resonance absorbance peak in the visible range of light, with its position being highly sensitive to the local refractive index surrounding the particle. Therefore, LSPR measures small changes in the wavelength of the absorbance position, rather than the angle as in traditional SPR [62].

PPI measurements through different methods are of complex interpretation. For example, analytical ultracentrifugation (AUC) enables longer periods of interaction between ligand and analyte molecules when compared to SPR or LSPR. This makes the data valuable but difficult to compare. As an example, epithelial cadherin (E-CAD) [63] a structural member of the adherens junction (AJ) family, has been studied using both SPR and AUC^63^. E-CAD is a calcium-dependent membrane protein with five extracellular immunoglobulin domains that plays a role in cell–cell adhesion [63]. In contrast to TJs, AJs do not encircle the entire perimeter of the cell and are found sporadically below the TJ. Taking advantage of the wealth of literature surrounding structural and functional studies of E-CAD, we used LSPR data for E-CAD as a control. To be consistent with our chimeric designs, we prepared MBP fusion to E-CAD (MBP-eCAD), see Appendix A. To validate our chimeric design, we first determined the affinity constant of MBP-eCAD. In the literature, E-CAD’s K_D_ is estimated to be in the range of 100 μM [63]. Through LSPR, we calculated the K_D_ of MBP-eCAD to be 197 μM, a 2-fold decrease in affinity. Upon closer inspection, a range of K_D_ between 100 and 200 μM was observed when K_D_ values were obtained by different techniques, AUC and SPR [63,64,65]. Considering our data estimated correctly the K_D_ of E-CAD, it follows to estimate that the K_D_ of MBP-CC1(Table 1) and, therefore, of CLDN1-CLDN1 homotypic interactions may be between 150 and 300 nM.

In Figure 2, we studied the homotypic interactions of MBP-CC1 and compared them to the homotypic interactions of MBP-eCAD. We observed that, when compared to MBP-eCAD, the homotypic interactions of MBP-CC1 were over 700-times higher, hinting at the relevance of CLDNs in the TJ, and the TJ itself as a keeper of the paracellular barrier in tissues. Although exogenous expression of OCLN does not result in TJ strand formation, OCLN is recruited to the TJ strands when co-expressed with CLDNs [13]. We tested the hypothesis that CLDN1 and OCLN may directly interact by creating a chimeric OCLN (MBP-COC), see Appendix A. SPR data presented in Figure 2 indicate that homotypic interactions of MBP-COC are 65-times less strong than MBP-CC1. The data suggest CLDN and OCLN may be part of different strands within the TJ.

Finally, we studied heterotypic interactions of MBP-CC1. CPE is reported to promiscuously bind a diverse number of CLDNs but not CLDN1 [39]. A mutant form of CPE (CPEm19) [66,67] was designed to bind CLDN1 with higher affinity [68]. We prepared MBP-CPE and MBP-CPE(m19), see Appendix A, in order to validate our findings with those available in the literature. Comparing homotypic and heterotypic interactions of MBP-CC1, our results indicate a rank of affinity, MBP-CC1 (homotypic) > MBP-CC1 vs. MBP-COC > MBP-CC1 vs. MBP-CPE(m19) > MBP-CC1 vs. MBP-CPE. This hierarchy seems to be in agreement with the literature stating that CPE(m19) has greater affinity for CLDN1 [68].

Finally, CPE(m19) [68] a broad-specific CLDN1 binder, decreased the paracellular but not transcellular integrity of treated epithelial cells. Taken together, MBP-CC1 may have an affinity for self in the nano molar range, while MBP-CC1 and CPE(m19) may have affinity in the low micromolar range.

### 2.3. Surface Plasmon Resonance (SPR) to Determine MBP-CC1 Structural Domains Responsible for Adhesion

Our bioSAXS data (Figure 1E) combined with the SEC chromatogram of MBP-CC1 (Figure 1D) revealed that purified MBP-CC1 forms a higher order oligomeric state that may describe the native protein structure in the plasma membrane microenvironment. We followed our experiments by mutating the two cysteines in the first extracellular loop of hCLDN1 in MBP-CC1 that are part of the signature of the CLDN family and are required for function [69,70,71]. Our results (Table 1) suggest that there is a loss of ~100-times affinity by the Ala mutation corresponding to hCLDN1 Cys54 and Cys64.

We proceeded to test through LSPR if decreasing the percentage of hCLDN1 in MBP-CC1 will also decrease the homotypic affinity (K_D_). The original design MBP-CC1, contained 50% of hCLDN1 amino acid sequence (Figure 1A). We prepared a 40%, and 30% chimeric hCLDN1(Appendix A), MBP-CC1 (40%) and MBP-CC1 (30%). As a final experiment, we deleted each extracellular loop in MBP-CC1 (30%), MBP-CC1 (30%ΔEL1) and MBP-CC1 (30%ΔEL2).

The MBP-CC1 (30%) retained the larger oligomerization displayed by MBP-CC1 50%, indistinguishable in the SEC chromatograms. Normalized K_D_, by the lowest affinity, that of MBP-CC1 (30%), shows how affinity increases with the increased content of hCLDN1 in the chimera (Table 1), while the oligomeric state remains almost unchanged. MBP-CC1 50% is ~3000-times stronger than MBP-CC1(30%), while MBP-CC1(40%) has a K_D_ 35-times higher than MBP-CC1 (30%). These results suggest that the participation of the transmembrane helices may be a significant contributor to the strength of the barrier while the extracellular loops may play a major role in assembly of the TJ and cell–cell adhesion. Our chimeric approach may be a good tool to solve some of the discrepancies from in silico experiments observed by Fuladi et al. in regard to Cis interactions of the transmembrane domains in CLDNs [26].

Finally, losing the entire first extracellular loop 1 (MBP-CC1 30%ΔEL1) results in a reduction in affinity of ~15,000 times (Table 1) when the K_D_ is normalized to that of MBP-CC1. A deletion of the second extracellular loop (MBP-CC1 30%ΔEL2) results in a decrease of ~1000-times affinity. SEC experiments show that MBP-CC1 30%ΔEL1 is a dimer, while MBP-CC1 30%ΔEL2 forms a dimeric and a tetrameric oligomer of equal area under the curve. Recent studies suggest that the first extracellular loop of CLDNs is sufficient to determine paracellular permeability [72]. Additionally, in the case of CLDN1, a peptide from the first extracellular loop (53–80) regulated paracellular gastric permeability in vivo [73]. On the other hand, the second extracellular loop of CLDNs may be a CPE binder [74]. Our data also converged to a similar conclusion, that MBP-CC1 homotypic interactions are greater than MBP-CC1 vs. MBP-CPE(m19).

### 2.4. MBP-CC1 In Vitro Experiments

The transepithelial electrical resistance (TEER) measurement is used to assess the barrier function of epithelial cells on porous supports [75]. Additionally, CLDNs play an important role in proliferation [4]. We measured TEER in Caco-2 cells and performed a proliferation assay on Cal27 cells [76]. Caco-2 cells are among the few cell lines that can register TEER values that permit the detections of changes due to agents that affect the paracellular space. The baseline TEER values of Caco-2 cell monolayers (on day 21 post-seeding) varied from 1200 to 1450 Ω·cm^2^. Caco-2 cells contain TJs composed of low levels of CLDN2, and high levels of CLDN1, CLDN4 and OCLN [77]. Important to note is that CLDN1 interacts with CLDN3 [6]. Figure 3A demonstrates how MBP-CC1, MBP-CC2 (see Appendix A), MBP-CC3 (see Appendix A) and MBP-COC affected TEER of 21-day post seeding cultures. MBP-CC2 had no effect on the recorded TEER values, while the other chimeric proteins did. Additionally, MBP-CPE also affected TEER perhaps due, in part, to its binding to CLDN4 [78].

CLDNs have been implicated in cell proliferation and tumor progression [79,80]. Tongue squamous cell carcinoma, Cal27 cells, contain one of the simplest TJs, composed of CLDN1, JAM-A, and OCLN, thus presenting a good model for the study of proliferation. We measured proliferation 24 h after cells were incubated with 1 μM protein (see Section 4). Loss of TJ-related adhesion between cells has been reported with malignant transformation and decreased proliferation [80,81]. In our experiment, we employed a number of controls. The addition of epithelial growth factor (EGF) increased proliferation of CAL27 cells 2 times. MBP-CC1 and MBP-COC decreased proliferation. MBP-CPE(m19), the CLDN1 binder, but not MBP-CPE, also decreased proliferation. MBP-CC1 exerted the biggest effect and it was statistically different than the control (no protein added). Taken together, the TEER and the proliferation assay seem to indicate that chimeric CLDNs may be suitable tools for the study of the TJ structure and function, with the added value that individual CLDNs may be targeted, as opposed to employing CPE or its mutants which are more promiscuous [68].

### 2.5. Zebrafish MBP-CC11A Effects In Vivo

TJs have been implicated in development [82], embryogenesis [19], and even birth defects [16]. Zebrafish is a model of choice to study development. The literature of CLDNs in teleost fish is extensive [83]. We selected CLDN11A from zebrafish, based on its expression in the gills and skin of the developing fish [83]. We thus created zebrafish chimeric CLDN11A (MBP-zfCC11A), see Appendix A. We incubated zebrafish embryos 12-h post-fertilization (hpf) with 10, 25 or 50 μg of protein in the embryo medium (1 mL). We imaged the embryos 21 hpf. Treatments included: no addition, MBP-2jua, MBP-zfCC11A. Figure 4A depicts embryos post treatment. Embryos have normal development when incubated at 28.5 °C, but their development is delayed when kept at room temperature. No treatment and addition of MBP-2jua (even at the highest concentration) resulted in normal development. We observed that MBP-zfCC11A had a concentration-dependent effect in delaying the embryos development (Figure 4A).

Somites are a set of bilaterally paired blocks of paraxial mesoderm that form in the embryonic stage of somitogenesis, along the head-to-tail axis in segmented animals (e.g., zebrafish). Figure 4, panel B, demonstrates how somites, and somites’ organization, is lost when treated with zfCC11A. Only the highest concentration treatment of MBP-2jua and MBP-zfCC11A is shown. A mesenchymal to epithelial transition (MET) defines the outer cellular “shell” of the developing somite, with the core cells remaining as a mesenchymal organization. CLDNs have been implicated in MET [81,84]. Body axis elongation represents a common and fundamental morphogenetic process in development. A key mechanism triggering body axis elongation without additional growth is convergent extension (CE), whereby a tissue undergoes simultaneous narrowing and extension [85]. Both collective cell migration and cell intercalation are thought to drive CE and are used to different degrees in various species as they elongate their body axis [85]. CLDNs also play a key role in CE [17].

As described above, development is a complex process, thus our goal was to demonstrate that chimeric proteins could be a tool for studying these processes in greater detail. Combining our results, we propose that chimeric CLDNs can interfere with cell organization, proliferation, CE, and somitogenesis in zebrafish. The scope of our experimentation is to provide a proof of concept for the use of chimeric CLDNs as described here. In addition, to provide evidence that it is possible to manipulate development without genetically modifications to the organism. In a recent study, Schwayer and colleagues [86], describe studies of Mechanosensation in zebrafish as a result of manipulation of the TJ. Alongside the extensive CLDN reports in zebrafish [83], our contribution can lead future research to understand molecular events associated with CLDNs and the TJ. Experimentation in chicken development can also employ the chimeric CLDN approach since it is also possible to manipulate the embryos ex ovo and study embryogenesis, organogenesis, limb development, etc. [18,22].

## 3. Conclusions

Chimeric CLDN proteins constitute a novel reagent for research and could be utilized in various translational applications such as the creation of molecules that can overcome blood–tissue barriers. We have presented evidence that chimeric CLDNs can be reagents for molecular and structural biology, biophysical applications, and can offer a close surrogate for CLDN’s structure and function. Our data, in agreement with the corresponding literature, indicate that chimeric CLDNs may be suitable tools for the study of development (embryogenesis, somitogenesis, organogenesis, etc.) without genetic manipulation of the model organism. With the evidence accumulated in this study, we suggest that the TJ as a unit may be organized, at the membrane level, in separate protein strands, CLDNs being one, and OCLN being another. We also suggest that the interplay between TJ and AJ (cadherins) may be regulated by other events than the direct interactions between CLDNs and cadherins.

## 4. Materials and Methods

### 4.1. Materials

All cloning and PCR reagents were obtained from New England Biolabs (NEB, Ipswich, MA, USA). Amylose resin was purchased from NEB and used according to manufacturer’s protocol. All chemicals were purchased from Sigma-Aldrich (https://www.sigmaaldrich.com/united-states.html) (accessed on 5 March 2020). pET28a empty vector was obtained from Sigma-Aldrich, catalog number 69864. pMAL c2x plasmid (discontinued from NEB) was used to generate maltose binding protein (MBP) for cytosolic expression as a gene of interest to clone into pET28a between restriction sites NcoI and NdeI (Supplementary Information).

### 4.2. Protein Expression and Purification

gBlocks for the chimeric proteins hCLDN1 (accession number O95832), hCLDN2 (accession number O15551), hCLDN3 (accession number O15551), hOCLN (accession number Q16625), enterotoxin *Clostridium perfringens* (CPE) (accession number CAA57443), mutant CPE(m19) [68] and E-CAD (Val102-Asp312, accession number P12830) were obtained from IDT DNA Technologies https://www.idtdna.com/pages (Supplementary Information) (accessed on 5 March 2020), codon optimized for *E. coli K-12* (IDT DNA Technologies Codon Optimization Tool). The gBlocks were amplified with forward and reverse primers, adapters of T7 promoter primer and T7 reverse primer, followed by restriction enzyme digestion (XhoI and NdeI). Fragments were subcloned in pET28a-MBP plasmid, kanamycin resistant (Supplementary Information). The final product produces an N-terminal MBP-fusion protein of the target with a C-terminal 6xHis tag. Cloning and subcloning transformations performed in NEB 5-alpha (NEB). Plasmids for protein expression were transformed into SHuffle T7 Express (NEB), spectinomycin resistant. Protein expression and purification (Amylose resin) were performed following manufacturer’s instructions. Eluate was concentrated by using Microsep Advance with 10k Omega centrifugal devices from Pall Corporation https://www.pall.com/ (accessed on 5 March 2020).

### 4.3. Protein Models

Models were created and visualized using UCSF chimera [87]. Maltose-binding protein (MBP) (PDB id: 5gxt), S-836 (PDB id: 2jua) and in silico model of hCLDN1 PDB file, generously shared by Dr. Shikha Nagidi (Department of Biomedical and Chemical Engineering, Syracuse University, Syracuse, NY, USA), were used to compose the models and graphic representations presented in Figure 1. Dr. Nagidi’s methods were previously described for other CLDNs [5,49,50].

### 4.4. Small-Angle X-ray Scattering for Biomolecules (bioSAXS)

SAXS data were collected at the Cornell High Energy Synchrotron Source (CHESS) G1 station in Ithaca, NY, USA. All experiments were performed at 21 °C. Protein samples (between 0.5 and 2 mg/mL) of MBP-CC1 were exposed with a 250 × 250 μm beam of 9.968 keV X-ray. Sample preparation included centrifugation at 30,000× *g* for 30 min and filtration to remove any aggregates. Samples (30 μL) were loaded and oscillated in the beam using an automated system with a plastic chip-based sample cell (2 mm path) and polystyrene X-ray transparent windows. The sample cell and X-ray flight path were placed under vacuum to reduce background scattering. Scattering patterns were captured on a Pilatus 100K-S detector (Dectris, Baden, Switzerland) at 1504 mm distance. The exposure time was 5 s for each image, and 10 images were recorded for each sample. All mathematical manipulations of the data (azimuthal integration, normalization, averaging and buffer subtraction), as well as error propagation, were carried out using RAW software49. The range of momentum transfer was calculated to be 0.0068 < q = 4π sin(θ)/λ < 0.28 Å^−1^, where 2θ is the scattering angle and λ = 1.257 Å is the X-ray wavelength. Samples were run at a range of concentrations (0.3, 0.6, 1.0, 2.0, 5.0, and 10 mg mL^−1^) to evaluate for possible concentration effects. Molecular weight estimated from a lysozyme standard (3.5 mg mL^−1^, 50 mM NaOAc, 50 mM NaCl pH 4.0) agreed with our expectations within error. The maximum dimension of the particle, D_max_, was estimated based on the goodness of the data fit and smoothness of the decaying tail. The GNOM output file for CC1 (50P) was used as input to DAMMIF35 to perform ab initio shape reconstruction without imposing any symmetry. The 20 reconstructed bead models were superimposed and averaged using DAMAVER in the automatic mode. The mean NSD was 0.536 ± 0.029 (*n* = 20), where an NSD value < 1 indicates close agreement between different reconstructed models.

### 4.5. Surface Plasmon Resonance (SPR)

Performed using Open SPR by Nicoya Lifesciences, https://nicoyalife.com/ (Canada) (accessed on 5 March 2020). We assayed protein–protein interactions by loading 0.100 mg of each protein as ligand into the Carboxy sensor chip (Nicoya Lifesciences). Following the blocking step (manufacturer’s buffer), 200 µL of 1 M sodium caprate was administered to disrupt the preformed protein–protein interactions. All proteins analyzed formed at least dimers, these species needed to be disrupted in order to determine new protein–protein interactions kinetics. Triplicate injections of the analyte protein in concentrations of 12.5 µg, 25 µg, 50 µg and 100 µg per 200 µL injections. Caprate injections were performed after each analyte interaction was concluded. The close curve fitting to the sensograms was calculated by global fitting curves (1:1 Langmuir binding model). The data were retrieved and analyzed with TraceDraw software (Kitchener, ON, Canada).

### 4.6. Tissue Culture

Colorectal adenocarcinoma epithelial cells (Caco-2, ATCC HTB-37) and tongue squamous cell carcinoma (Cal27, ATCC CRL-2095) were obtained from American Type Culture Collection (https://www.atcc.org/) (accessed on 5 March 2020) and cultured according to the guidelines provided by the organization.

### 4.7. Trans Epithelial Electrical Resistance (TEER)

Transepithelial electrical resistance (TEER) measurements were made using a Millicell-ERS device (Millipore, Burlington, MA, USA) and chopstick-style electrodes. Briefly, Caco-2 cells were used between passage numbers 29–33. Cells were seeded on to Millipore Millicell^®^ cell culture inserts (0.4 μm pore size) in 24-well plates at 1 × 10^5^ cells/cm^2^. Caco-2 monolayer formation in transwells was assessed by measuring TEER using a Millicell-ERS device (Millipore) and chopstick-style electrodes. Growth medium was removed, and the differentiating monolayers were gently washed twice with Hanks Balanced Salt solution (HBSS) and finally placed in 400 µL of HBSS. The growth medium was also removed from the basolateral chamber and replaced with 750 µL of HBSS. Measurements were made at room temperature (25 °C). Proteins were added to a final concentration of 1 μM. Replicates of four separate determinations are reported.

### 4.8. Proliferation Assay

Using Cal27 cells we performed proliferation assays using ATPlite Luminescence Assay System (https://www.perkinelmer.com/) (accessed on 5 March 2020) following manufacturer’s instructions. Protein was added to a final concentration of 1 μM. Human epithelial growth factor (EGF) was purchased from Abcam (https://www.abcam.com/) (accessed on 5 March 2020) and used at a final concentration of 10 ng/mL.

### 4.9. Zebrafish

#### 4.9.1. Transgenic Line

Zebrafish transgenic line with ubiquitous expression of membrane-bound GFP (Tg(*bactin2*:EGFPCAAX)^z200^) was obtained from Kristen M. Kwan. Construction of the transgenic line is described in a recent publication [88]. Transgenic zebrafish were mated with wildtype lines to produce experimental embryos. Embryos were screened for expression of GFP phenotype.

Dechorionation: Embryos were divided into new plates—60 embryos per plate, with fresh embryo medium and dechorionated 8 h post-fertilization with forceps.

Protein treatment: Dechorionated embryos were placed in wells using a heat-sterilized wide-bore glass pipette 12 h post-fertilization. Four embryos were placed per well, excess embryo medium removed, and 1 mL of fresh embryo medium was added gently on top of embryos. In total, 10 μg, 25 μg, or 50 μg of concentrated chimera claudin protein (zclaudin- zclaudin11A) was added to embryos in 1 mL of embryo medium. Plate was wrapped in parafilm and placed in an incubator at 28.5 °C for 20 h. Room temperature treatment was wrapped in parafilm and left on the counter for 20 h.

#### 4.9.2. Zebrafish Microscopy

Images were acquired using the 20× lens at 2× zoom and the 4× lens at 1.5× zoom on an Olympus FV1000 confocal microscope. A Kalman filter by line was used, and images were taken sequentially by line when the RFP and GFP fluorescence were both present. A 7% 405 laser with TD1 for brightfield, a 10.1% 488 laser with EFGP for GFP, and a 10.1% 546 laser with Alexa Fluor 546 for RFP fluorescence were used. Images were taken in slices (depth) and compiled with ImageJ. The frame of the 20×/2× zoom images was approximately 152.78 × 152.78 microns taken with 640 × 640 resolution. The 20× lens had an NA of 0.75. Each 20× z-stack had around 30 slices. The frame of the 4×/1.5× zoom images was approximately 2342.40 × 1873.92 microns taken with 800 × 640 resolution. The 4× lens had an NA of 0.13. Each 4× z-stack had around 10 slices or was taken as a single slice. HV and laser levels were kept consistent for accurate qualitative comparison of fluorescence between treatment conditions.

## Figures and Tables

**Figure 1 ijms-22-04947-f001:**
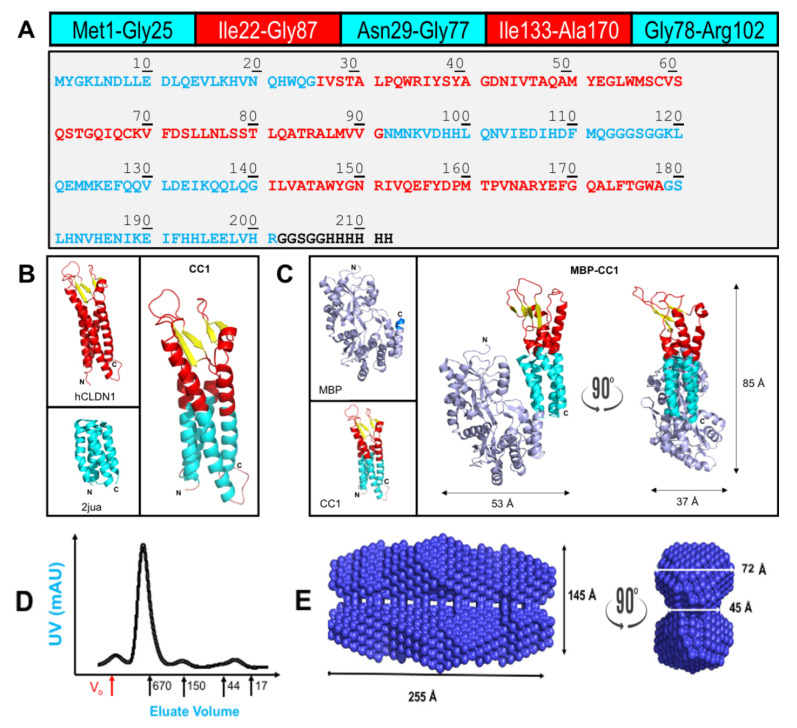
Synthetic design and characterization of MBP-CC1. (**A**) Amino acid composition of CC1. Light blue is soluble protein 2jua and in red is hCLDN1. All chimeric constructs have MBP as N-terminal fusion and a 6xHis tag at the C-terminus. Additionally, we present the full amino acid sequence of the translated CC1. (**B**) Graphical representation of the chimeric design to produce CC1. (**C**) Graphical representation of the chimeric design to produce MBP-CC1. The dimensions of relevant axis of MBP-CC1 are presented. MBP, in light blue, also displays, in dark blue, the position and structure of the short linker of four amino acids (Asn, Ala, Ala, Ala). (**D**) Size-exclusion chromatography of MBP-CC1. The monomeric MBP-CC1 has a molecular weight of 65 kDa. When comparing its elution volume with the protein standards (BioRad Gel Filtration Standard, cat. Number 1511901), MBP-CC1 appears to elute above 670 kDa. Below the x-axis, V_o_ is void volume, and the numbers correspond to standards of molecular weight. (**E**) Small-angle X-ray scattering (bioSAXS) of MBP-CC1 performed at 21 °C. Two different orientations of the volumetric data are presented. bioSAXS data collected for MBP-CC1resulted in a radius of gyration (R_g_) of 83.1 ± 4.03, and a maximum particle size (D_max_) of 305 Å.

**Figure 2 ijms-22-04947-f002:**
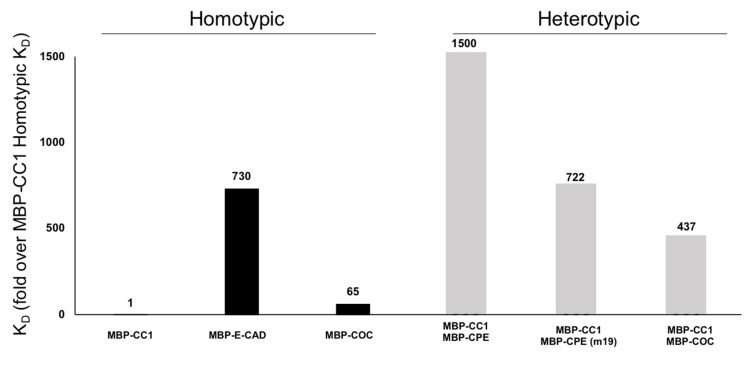
Surface Plasmon Resonance (SPR) of MBP-CC1. The constant of affinity (K_D_) of several proteins associated with the TJ were determined by SPR. Using MBP-CC1 homotypic interaction as a control, we normalized the K_D_ of the other interactions. Homotypic interactions of MBP-CC1, MBP-eCAD (AJ), and MBP-COC. These results suggest that CLDN1 interactions are 700-times stronger than MBP-eCAD or 65-times stronger than MBP-COC. We followed this analysis with heterotypic interactions of MBP-CC1 with MBP-CPE and MBP-CPE(m19) mutant or the interactions between MBP-CC1 and MBP-COC.

**Figure 3 ijms-22-04947-f003:**
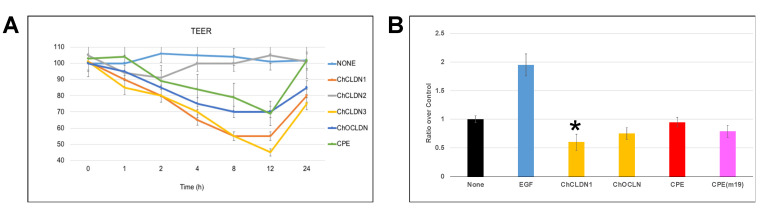
MBP-CC1 in vitro experiments. (**A**) Caco-2 cells were treated with MBP-CC1, MBP-CC2, MBP-CC3 or MBP-COC. No addition of proteins or MBP-CPE was used for controls. The graph represents the change in TEER compared to the control (no addition of protein). Following treatment, TEER was monitored for 24 h (intervals of 0, 1, 2, 4, 8, 12, and 24 h). Three experiments are averaged in the graph (±SD). (**B**) Cal27 cells were treated with EGF, MBP-CC1, MBP-COC, MBP-CPE or MBP-CPE(m19). After 24 h of treatment, cells were prepared for the proliferation assay (ATPlite, Perkin Elmer). Proliferation is reported as the average of 4 separate experiments (±SD). For statistical analysis, we employed t-test, and the asterisk represents a statistical significance (*p* < 0.001).

**Figure 4 ijms-22-04947-f004:**
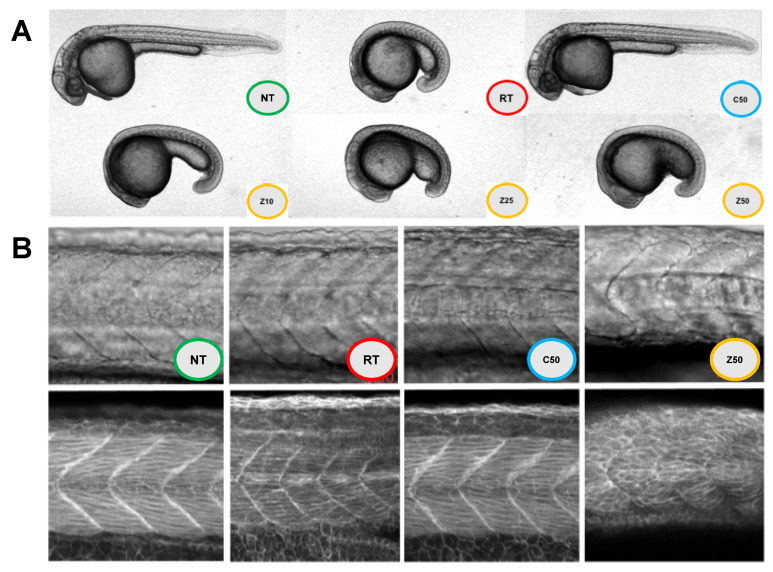
MBP-zfCC11A, in vivo. MBP-zfCC11A was the reagent selected to validate the use of chimeric CLDNs to observe its effects in vivo. Zebrafish embryos treated at 12 hpf, imaged at 21 hpf. Images taken with 20× lens at 3× zoom. Panel (**A**), Zebrafish embryos. **NT**, no treatment; **RT**, embryos maintained at room temperature; **C50**, 50 μg of MBP-2jua as control; **Z10, Z25, Z50**, 10, 25 or 50 μg/mL of MBP-zfCC11A. Panel (**B**), qualitative analysis of somite development. **NT**, no treatment; **RT**, embryos maintained at room temperature; **C50**, 50 μg of 2jua as control; **Z50**, 50 μg of zfCC11A.

**Table 1 ijms-22-04947-t001:** Surface Plasmon Resonance (SPR) studies of chimeric claudins and other relevant proteins. (*) MBP-eCDA experiments are carried out in the presence of 3 mM CaCl.

PPI Evaluated	K_a_ (1/(M*s))	K_d_ (1/s)	K_D_ (M)
MBP-eCAD vs. MBP-eCAD *	4.65 × 10^3^ ± 1.61 × 10^2^	6.96 × 10^−4^ ± 7.87 × 10^−5^	1.97 × 10^−7^ ± 2.25 × 10^−8^
MBP-CC1 vs. MBP-CC1	8.88 × 10^4^ ± 4.11 × 10^3^	2.42 × 10^−5^ ± 1.26 × 10^−6^	2.70 × 10^−10^ ± 2.45 × 10^−11^
MBP-CC1 vs. MBP-CC1 (50%ΔC)	1.47 × 10^3^ ± 1.05 × 10^1^	5.30 × 10^−5^ ± 1.78 × 10^−6^	3.60 × 10^−8^ ± 3.76 × 10^−9^
MBP-CC1 vs. MBP-CC1 (40%)	3.92 × 10^3^ ± 2.02 × 10^1^	8.90 × 10^−5^ ± 5.71 × 10^−7^	2.27 × 10^−8^ ± 2.63 × 10^−10^
MBP-CC1 vs. MBP-CC1 (30%)	6.60 × 10^3^ ± 0.29 × 10^1^	4.98 × 10^−3^ ± 8.89 × 10^−8^	7.54 × 10^−7^ ± 3.36 × 10^−8^
MBP-CC1 vs. MBP-CC1 (30%ΔEL1)	5.35 × 10^2^ ± 7.43 × 10^1^	2.13 × 10^−3^ ± 8.17 × 10^−6^	3.98 × 10^−6^ ± 5.79 × 10^−7^
MBP-CC1 vs. MBP-CC1 (30%ΔEL2)	1.32 × 10^3^ ± 2.31 × 10^1^	3.80 × 10^−4^ ± 1.95 × 10^−6^	2.87 × 10^−7^ ± 6.49 × 10^−9^
MBP-CC1 vs. MBP-CPE	2.06 × 10^3^ ± 1.71 × 10^2^	8.71 × 10^−4^ ± 2.09 × 10^−5^	4.22 × 10^−7^ ± 4.53 × 10^−8^
MBP-CC1 vs. MBP-CPE (m19)	9.64 × 10^3^ ± 2.41 × 10^2^	1.88 × 10^−3^ ± 4.86 × 10^−6^	1.95 × 10^−7^ ± 5.39 × 10^−9^
MBP-CC1 vs. MBP−2jua	1.31 × 10^2^ ± 2.12 × 10^1^	1.80 × 10^−1^ ± 6.84 × 10^−2^	1.38 × 10^−4^ ± 2.34 × 10^−5^
MBP-COC vs. MBP-COC	3.67 × 10^3^ ± 4.17 × 10^1^	6.47 × 10^−5^ ± 3.39 × 10^−6^	1.76 × 10^−8^ ± 1.13 × 10^−9^
MBP-CC1 vs. MBP-COC	2.09 × 10^3^ ± 2.61 × 10^1^	2.48 × 10^−4^ ± 1.11 × 10^−5^	1.18 × 10^−7^ ± 6.27 × 10^−9^

## Data Availability

The data that support the findings of this study are available from the corresponding author upon reasonable request.

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
