# Peer review of "Chimeric Claudins: A New Tool to Study Tight Junction Structure and Function"

_ijms, 2021, doi:10.3390/ijms22094947_

Round 1

Reviewer 1 Report

In this study, various chimeric claudins were prepared and studied. The aims of this study are quite unclear and the authors should provide a better explanation of the purpose of these experiments and connect more clearly the functional data with the structural aspects of the chimeric claudins. L. 60-67 need more elaboration.

All constructs appear to have MBP. Please clarify that better in Fig. 1 and the data obtained with SAXS. Consider also using the abbreviation MBP-CC1. 

Unfortunately, the authors in this study opted to preserve MBP in the construct.  However,  I believe they could get cleaved protein and subsequently remove the uncleaved+free MBP with affinity chromatography. Their results would have therefore been more clear and without any questions about interference from MBP (ideally the His-tag should have also been removed).

It would be good if the crystal packing in the structures of various claudins is examined and if the way that molecules are packed in the crystal lattice shows any resemblance to the oligomerization pattern. 

Some minor correction and suggestions:

  1. Please use 'times' instead of 'fold'
  2. Consider using CLDNs when you refer to plural (claudins)
  3. l.24... is 'the' most apical
  4. l.28 ...and 'has' been found (you refer to TJ I suppose)
  5. l.56 ...could have been 'preserved'
  6. l.100 'size-exclusion chromatography'
  7. Provide temperature of the SAXS experiments

Author Response

Estimated Reviewer

We received with great enthusiasm your comments regarding our manuscript.

We made a greater effort to support through the literature the motivation of our work. Proteins that through detergent-based methods cannot be studied structurally can be successfully prepared as chimeric proteins. We explained the use of MBP and as per your request renamed all our constructs as “MBP-xx” thus each control and each sample are clearly recognized in their nature.

We also explain that the strong adhesive nature of claudins make it difficult to obtain a cleaved product. MBP-TEV-CC1 rendered mixed products, with cleaved and non-cleaved proteins remaining in the purified fraction. This would have affected our ability to conduct SPR, a key piece of evidence regarding the strength of the interactions offered by CLDNs in the native TJ.

We also provide literature to support the lack of information regarding Cis- and Trans- CLDN-CLDN interactions derived from Crystal Packing. We also provide literature evidence that even in silico models relying on the crystal structures of CLDNs are contradicting regarding these interactions due to the lack of better structural information.

We improved the correct labeling of the axis in the graphs, nomenclature, and overall clarification.

Due to the scope of our work, to present a synthetic biology approach (protein engineering) as a solution to structural and functional studies of CLDNs we mainly focused on that aspect. We provide sufficient details of the construction of the chimeras, details of the structural and biophysical studies. Additionally, we present three proofs of concept regarding the usefulness of the chimeric CLDNs. In vivo we show these reagents can be used to study paracellular permeability and proliferation. In vivo we present evidence for studies of development using the model organism Zebrafish. We plan future studies to focus only on development and other studies focused only on the molecular networking of the TJ through CLDNs, their effect in cell proliferation is just the proof of concept that matches the reports in the literature, as is the TEER experiments. New tools have been published to study mechanotransduction associated with the TJ in Zebrafish, we cite the work and leave for a future study with more in-depth experimentation.

We thus thank you for your critical evaluation of our work that made us prepare a stronger manuscript for publication.

Sincerely

Dario Mizrachi, Ph.D. and the team of co-authors.

Reviewer 2 Report

In this work, authors have presented chimeric claudins as a new tool for studying tight junctions. The work is interesting, however, some points need to be addressed.

  1. The introduction is very short and needs a more details about other methods of studying tight junctions in literature. Authors should also provide advantages and disadvantages of these approaches.
  2. Authors should mention that they have validated their work in zebra fish embryo and Caco-2 cells.
  3. Authors should provide quantitative data for protein structure analysis of the chimera. 
  4. Did the authors perform analysis of how the length of the resulting chimeric protein affects the TJ complex functionality and downstream signalling?
  5. Plots should have proper axes labelling. For example: fig3a does not have a y-axis label.
  6. The development of zebrafish embryos should be quantified for example Body length over time and somite patterning.

Author Response

Estimated Reviewer

We received with great enthusiasm your comments regarding our manuscript.

We made a greater effort to support through the literature the motivation of our work. Proteins that through detergent-based methods cannot be studied structurally can be successfully prepared as chimeric proteins. We explained the use of MBP and as per your request renamed all our constructs as “MBP-xx” thus each control and each sample are clearly recognized in their nature.

We also explain that the strong adhesive nature of claudins make it difficult to obtain a cleaved product. MBP-TEV-CC1 rendered mixed products, with cleaved and non-cleaved proteins remaining in the purified fraction. This would have affected our ability to conduct SPR, a key piece of evidence regarding the strength of the interactions offered by CLDNs in the native TJ.

We also provide literature to support the lack of information regarding Cis- and Trans- CLDN-CLDN interactions derived from Crystal Packing. We also provide literature evidence that even in silico models relying on the crystal structures of CLDNs are contradicting regarding these interactions due to the lack of better structural information.

We improved the correct labeling of the axis in the graphs, nomenclature, and overall clarification. Better characterization of the protein and details of the small-angle x-ray scattering experiment.

Due to the scope of our work, to present a synthetic biology approach (protein engineering) as a solution to structural and functional studies of CLDNs we mainly focused on that aspect. We provide sufficient details of the construction of the chimeras, details of the structural and biophysical studies. Additionally, we present three proofs of concept regarding the usefulness of the chimeric CLDNs. In vivo we show these reagents can be used to study paracellular permeability and proliferation. In vivo we present evidence for studies of development using the model organism Zebrafish. We plan future studies to focus only on development and other studies focused only on the molecular networking of the TJ through CLDNs, their effect in cell proliferation is just the proof of concept that matches the reports in the literature, as is the TEER experiments. New tools have been published to study mechanotransduction associated with the TJ in Zebrafish, we cite the work and leave for a future study with more in-depth experimentation.

We thus thank you for your critical evaluation of our work that made us prepare a stronger manuscript for publication.

Sincerely

Dario Mizrachi, Ph.D. and the team of co-authors.

Round 2

Reviewer 1 Report

I appreciate the new corrections and clarifications made by the authors. I still feel that the legend in Fig. 1  is misleading and you should also change CC1 to MBP-CC1. Please also correct 'The monomer CC1, 65 kDa, appears to elute above 670 kDa' - the sentence doesn't make any sense (the protein elutes at 670 kDa suggesting the presence of decamers).

Author Response

Our team of researchers is very grateful to all your comments.

Regarding the second round of comments, we addressed the legend of Figure 1 to reflect a clear explanation of the conclusion that MBP-CC1 may form decamers based on the comparison with Molecular Weight Standards.

We carefully reviewed the manuscript looking for mismatches of the nomenclature we have adopted after your first-round review.

That mostly affected the names of constructs in Table 1. Now addressed.

Thanks 

Dario Mizrachi

Reviewer 2 Report

Authors have clubbed the response to both the reviewers comments in a single response. Authors should provide point by point response for each reviewer's comments separately and the manuscript should have the changes marked in red as it is hard to tell what changes were made. Authors should not expect reviewers to search for changes.

Authors do not seem to have addressed appropriately the comments regarding the introduction and the comments regarding the quantitative analysis of chimera structure. 

Author Response

We appreciate your second round of comments.

We assure you our goal was to show to both reviewers who had similar concerns about the changes we have made. Both letters had similar comments but were not identical. I do apologize.

We are attaching a version of the manuscript that has color-coded changes. Red are related to the first round of review and blue reflects the comments of the second round.

  1. We extended the introduction and identified a more logical step-by-step manner to explain the design of the chimeric work and of the experiments. We have used cited to explain the limitations of the chimeric work, and where these are mostly used. We suggest our approach presented here may be considered as an extension to the chimeric approach. (The introduction is very short and needs more details about other methods of studying tight junctions in literature. Authors should also provide advantages and disadvantages of these approaches.)
  2. We clearly highlighted the in vitro and in vivo experiments with sufficient citations to compare our results.
  3. We have expanded the structural characterization of the constructs. We discuss the formation of CC1 as well as MBP-CC1. We present models, and how they were derived, to explain and make sense of the bioSAXS models we display in Figure 1.E. We also report on the radius of gyration and maximum size of particles (Dmax) in the bioSAXS experiment.
  4. We have addressed the lack of adequate labeling of the axis in Figure 2 and Figure 3. We also reviewed the names of the constructs in Table 1 and carefully matched them with the nomenclature adopted after the first round of reviews.
  5. The main goal of this article is to identify a detergent-independent solution for structural and functional studies of the TJ membrane components. We used in vivo, and in vivo experimentation to demonstrate the usefulness of the method. In vivo experiments, we have followed trends in which rather than reporting measurements of the size of the embryos researchers display the embryos side-by-side while reporting that the images were captured under the same exact magnification.
  6. Future work will use chimeric CLDNs to study molecular events regulated by these proteins. We are currently studying birth defects using chick embryos. We have prepared chimeric CLDNs responsible for neural tube defects. The studies we cite were done with CPE, which is promiscuous. We are using individual chimeric CLDNs to pinpoint their function. Additionally we will move deep inside the cellular structures looking for answers. Chick embryos are a great model and in a way are better than Zebrafish. Zebrafish contains over 50 CLDNs making difficult the interpretation of some results due to the promiscuity of these proteins and adhesive properties.
